# Biotech-Educated Platelets: Beyond Tissue Regeneration 2.0

**DOI:** 10.3390/ijms21176061

**Published:** 2020-08-23

**Authors:** Sheila Siqueira Andrade, Alessandra Valéria de Sousa Faria, Manoel João Batista C. Girão, Gwenny M. Fuhler, Maikel P. Peppelenbosch, Carmen V. Ferreira-Halder

**Affiliations:** 1PlateInnove Biotechnology, Piracicaba, SP 13414-018, Brazil; 2Department of Biochemistry and Tissue Biology, University of Campinas, UNICAMP, Campinas, SP 13083-862, Brazil; alessandravsfaria@gmail.com (A.V.d.S.F.); carmenv@unicamp.br (C.V.F.-H.); 3Department of Gastroenterology and Hepatology Medical Center Rotterdam, NL-3000 CA Rotterdam, The Netherlands; g.fuhler@erasmusmc.nl (G.M.F.); m.peppelenbosch@erasmusmc.nl (M.P.P.); 4Department of Gynecology, Federal University of São Paulo, UNIFESP, São Paulo, SP 04024-002, Brazil; girao@unifesp.br

**Keywords:** platelets, biotechnology, tissue regeneration, biomaterials, growth factors

## Abstract

The increasing discoveries regarding the biology and functions of platelets in the last decade undoubtedly show that these cells are one of the most biotechnological human cells. This review summarizes new advances in platelet biology, functions, and new concepts of biotech-educated platelets that connect advanced biomimetic science to platelet-based additive manufacturing for tissue regeneration. As highly responsive and secretory cells, platelets could be explored to develop solutions that alter injured microenvironments through platelet-based synthetic biomaterials with instructive extracellular cues for morphogenesis in tissue engineering beyond tissue regeneration 2.0.

## 1. Introduction

### Biotech-Educated Platelets (Concept and Context)

Biotechnology is going to govern the actual decade of innovation at research institutions, as well as the industry 4.0. The use of living systems and organisms to identify and develop products has increased dramatically in recent years. Indeed, in the last 60 years, the sheer volume of knowledge about human biology has doubled every 10 years, raising awareness of the concept of biomimetics and its applicability. In this new decade, we will add therapies that can restore lost functions in ailing tissues and organs to the arsenal of health-aging biotechnology. Materials like “bio-skin/second skin” fabric that peels back in reaction to sweat and humidity [1,2]; cellular therapies, including the transplantation of stem cells and genetically manipulated cells to aid the repair of damaged or diseased tissue [3,4]; trends on blood cells (platelets) production using bioreactors that mimic the conditions inside the bone marrow [5,6,7,8,9], to name a few examples. Research is currently entering a collaborative endeavor where the natural and the artificial have blended and are no longer in orthogonal paths. In the context of cell-based tissue repair, our ability to restore damaged tissues and organs today relies mainly on two types of human cells that appear at the core of regenerative medicine: (1) nucleated mesenchymal stem cells and (2) anucleate platelets. The increasing discoveries in the field of platelet biology and its functions in the last decade undoubtedly show that platelets are one of the most biotechnological human cells. Forty years ago, researchers were convinced that platelets were merely “inanimate” cell fragments, only involved in the formation of a platelet plug to arrest bleeding [10,11]. Once platelets were primarily known for their role in hemostasis and thrombosis, several discoveries revealed that platelets had been increasingly recognized as multipurpose cells, with nuclear emancipation [12]. Despite their fragmentary ontogenesis (lacking nuclei and genomic DNA) from megakaryocytes, platelets are now considered versatile cells that continue to regulate protein generation after the fragmentation of megakaryocytes [13,14,15]. Indeed, they can influence a wide range of seemingly unrelated (patho-) physiological events like hemostasis, lymphogenesis, angiogenesis, tissue regeneration, infection, immunity, cancer, and the genesis and progression of cardiovascular diseases [16,17,18].

Every milliliter of human blood contains hundreds of millions (150,000–400,000/µL) of platelets; they detect blood flow patterns and check for signals derived from blood vessel walls, and their proximity to endothelial cells promotes the recognition of vascular wall lesions or wounds [17,19]. Platelets can “check and announce” with each other and with other cells via a range of bioactive substances secreted from their intracellular granules, and they synthesize their own proteins when in critical situations [10,20]. Hence, in this review, we focus on new concepts in the context of these attractive potential platelet features: “Biotech-educated platelets” that are conceptual and technological innovations; these new concepts assign versatile biotechnological roles to platelets as a central scientific system with molecular and biomimetic bases. “Biotech-educated platelets” are, therefore, considered platelet-based products for therapeutic use and experimental insights into translating platelet technology to the clinic and market. Some platelet discoveries have emerged in the current global scenario of biotechnological innovations, such as (1) serum-free platelet media supplement—a human platelet lysate for mesenchymal stem cell expansion for future application in regenerative medicine and an attractive option for commercial therapeutic development; (2) platelet membrane cloaking of functional nanoparticles (bio interfacing) for drug delivery [21]; (3) the production of biomimetic nanorobots consisting of gold nanowires cloaked with a hybrid of red blood cell membranes and platelet membranes for the removal of pathogenic bacteria and toxins [22]; (4) lyophilization of platelets stabilized with paraformaldehyde for future applications [23]; (5) construction of synthetic platelet-mimetic analogs designed to promote aggregation and hemostasis [24,25]; (6) generation of non-donor-derived platelets based on “bioreactor-on-a-chip” technology [7,25,26,27,28].

Looking at the clinical use of platelets, platelet transfusions are being administered either as a prophylactic to minimize the risk of bleeding or as a therapeutic strategy to control bleeding, including cases of hypo-proliferative thrombocytopenia (e.g., post-myelosuppressive chemotherapy). The current platelet demand exceeds supply by ~20%, while simultaneously, in many centers, packed platelets are amongst the most often discarded blood products due to expiration dates for transfusion that fall within 5–7 days from the storage date (according to clinical practice guidelines in transfusion medicine) [29]. Since 2015, in some blood centers of Europe and the USA, outdated platelet concentrates are frozen and stored for the further manufacturing of human platelet lysate (HPL) to support the ex vivo propagation of mesenchymal stem/progenitor cells (MSC) applied to tissue regeneration, a cell-based medicine. PlateInnove Biotechnology, a Brazilian startup, began addressing the need to develop humanized platelet-based culture assays with a platelet-based gel that improves micro-vascularization in a complete 3D cell culture support or scaffold by exploring the platelet content following the biotech-educated platelet context. Thus, we pursue this goal by establishing a scalable and good manufacturing practice (cGMP) that is compliant with the commercial platforms for scientific research and aligned with the guidelines of the Brazilian National Health Surveillance Agency (in Portuguese: Agência Nacional de Vigilância Sanitária, ANVISA). We are starting to build up a platelet platform in which the platelet molecular signature towards different stimuli can be explored in order to guide a rational production of a broad spectrum of biomolecules. Our approach uses big data analytics based on OMICS such as proteomics, peptidomics, secretomics, and sheddomics, which will be templates for the in silico designs of biomimetics platelet-products. In this investigation, we identified several growth factors, cytokines, chemokines, proteolytic enzymes, and matrix proteins/adhesion molecules, confirming what is already described in the literature and identifying some previously unknown proteins in the platelet content [30,31,32]. These results allowed for the identification of potential markets for platelets and their products. For example, platelet growth factors can be used for cell culture as mentioned, tissue regeneration, wound healing, drug delivery, and even skin rejuvenation in the cosmetics industry [33,34]. Our knowledge of platelets has increased considerably; we have learned about platelet functions and its unexpected biochemistry, especially in regenerative medicine. Patients around the world are waiting for therapies that demand biomaterials and cell transplantation, and companies that are creating and developing these innovations are spending billions of dollars. The Food and Drug Administration (FDA) in the US, and even ANVISA in Brazil (beginning in 2020), are starting to regulate therapies and products on a regular basis, granting approval and licensing to these developers to operate. Thus, in this review, we describe the potential implications of Biotech-educated platelets in the context of platelet-regenerative properties, including synthetic biomaterials based on platelet content. Here, we provide information on how optimized Biotech-educated platelets (specific platelet degranulation in a spatial and temporal dependent growth factor gradient combination) might affect dynamic processes such as hemostasis, angiogenesis, and wound healing, beyond tissue regeneration.

## 2. The Unexpected Biochemistry of Platelets Can Orchestrate the Entire Process of Wound Repair and Tissue Regeneration

Platelets resemble simple cells, including their biochemistry machinery (initial just for protein storage) [11,35]. However, over the years, the molecular processes that control platelet functions in hemostasis and thrombosis have begun to show results indicating the presence of several new mechanisms and new proteins, previously not expected, at least in the light of known and conventional concepts. Although it sounds repetitive, knowledge about the nuclear emancipation of platelets is recognized as a tremendous landmark in platelet biology evolution, which we cannot fail to highlight in this review: (1) platelets possess a substantial and diverse transcriptome, their transcription factors act in the platelet activation process and can also be secreted from activated platelets in microparticles, therefore, modulating gene expression in target cells [36]; (2) platelets keep functional spliceosome components to process mRNA [13]; (3) platelets have a signal-dependent protein translation with the presence of the platelet-derived extracellular ERp57, a thiol isomerase enzyme with an essential role in ensuring the correct folding of proteins before they are secreted to the cell surface or beyond [37]; (4) platelets possess the components of the microRNA processing system [15,37,38,39]; (5) platelets’ lifespans could be mediated by apoptosis and the desialylation process, independent of the intrinsic apoptotic pathway (Bak/Bax-mediated mitochondrial outer membrane permeabilization) [39,40].

In summary, these diverse observations attest that platelets are healers that produce growth factors and other bioactive molecules (soothing messengers), helping damaged tissue to rebuild. Platelets are “sentinel soldiers” that spark the protective response to inflammation, alert immune cells, and even attack microbial interlopers [10]. In fact, platelets are long-haul cell-vehicles for drug delivery, which pick up and deliver chemicals such as serotonin that helps the liver to regenerate after injury. Different drug delivery systems have been described for use in clinical practice, such as liposomes, nanoparticles, polymeric micelles, and others. However, the use of these systems has limitations ranging from low stabilities to low circulation times in vivo [41]. Cells or membranes derived from cells such as red blood cells, leukocytes, platelets, and stem cells have been extensively studied as a promising strategy aimed at minimizing the interaction with normal cells, reaching target cells, and releasing the drug in a controlled manner. In addition, these systems exhibit an excellent biocompatibility and biodegradability and are able to evade the immune response [42,43]. In this scenario, platelets have attracted attention because the biocompatibility of these cells is superior compared to other carriers, they have a life span of 7 to 10 days in vivo, and show a clearance similar to natural platelets when encapsulated with drugs [34]. Specifically, concerning cancer, tumor cells have the ability to induce platelet aggregation, culminating with the adhesion of platelets to the tumor. This effect could contribute to a targeted delivery if the activated platelets releasing granules would also release a drug [21,44].

The capacity to heal and restore ourselves from minor damage to the devastating injuries caused by diseases through regenerating restored tissue is closely associated with the multiple powers and plasticity of platelets, as mentioned above. As might be predicted, platelets are even “architects and engineers”, shaping the vascular system in wound healing processes from the newborn life stage to adulthood [17,45,46]. In fact, we heal by doing two things: (1) clotting—blood platelets have the capacity to create a clot/plug and stop bleeding, and as a consequence we experience (2) inflammation—our body (with the participation of platelets) chooses to replace damaged tissue, either in the presence of disease and/or some sort of injury, by creating scars based on fibrotic tissue. This ability to respond to injury and tissue repair is a fundamental property of all multicellular organisms, also considered the most complex biological process that occurs in human life. After an injury, multiple biological pathways immediately become activated and synchronized to respond. In all of these processes, platelets are “first responders” and protagonists [19]. Platelets start acting immediately after tissue damage, releasing components in the coagulation cascade, inflammatory mediators, and immune system players—all needed to prevent ongoing blood and fluid losses, also to remove dead and devitalized (dying) tissues to prevent infection. Hemostasis is achieved initially by platelets at work with the formation of a platelet plug followed by a fibrin matrix, which becomes the scaffold for infiltrating cells. When looking at the injury microenvironmental cues, the endothelial damage exposes the von Willebrand factor (vWF), subendothelial matrix, proteins, and collagen. A thrombus formation is triggered by platelet adhesion to collagen (via glycoprotein VI (GPVI) receptors) and vWF (via glycoprotein (GP) Ib–IX–V complex) [47,48,49]. In aggregating platelets, the binding of fibrinogen to the αIIbβ3-activated integrin triggers aggregation, while the vascular tissue factor (TF) triggers the initial generation of thrombin. Thrombin stimulates the formation of procoagulant (phosphatidylserine-exposing) and coated (fibrin-binding) platelets, which leads to the amplification of thrombin generation at procoagulant platelets; full-secretory platelets express P-selectin and CD63 glycoproteins and lysosomal-associated membrane protein 1 (LAMP1) on the outer membrane. The activated platelets release mediators, particularly adenosine 5′-diphosphate (ADP), and thromboxane A2 (TXA2), attracting circulating platelets to the growing thrombus [50]. Unexpectedly, in critical situations, platelets start synthesizing their own proteins to control this process, such as plasminogen activator inhibitor-1 [51], interleukin (IL)-1b [36], cyclooxygenase-1 (COX-1) [52], B cell lymphoma-3 (Bcl-3) [53], tissue factor [54], TIMP2 [14], and P-selectin [55]. Platelet heterogeneity can have major consequences for platelet functions [20], and it may be one of the limiting factors for effective tissue repair. In recent years, the concept of platelet structural heterogeneity in terms of size (small or large), content (light or dense, with or without secretion, with or without activated integrins), protein expression profile (low or high signaling, glycoprotein content, fibrin coat, or agonists), and age (young platelets with membrane ballooning or old platelets with exposed phosphatidylserine) has gained renewed interest [20,45]. The heterogeneity of thrombus compositions is also promoted by extrinsic (environmental) factors such as blood flow dynamics, vascular environment, and the local availability of platelet agonists. Environmental changes, as a consequence of disease, can affect the contents of platelets and their population and responses, for instance, by positive priming. In addition, platelets store and release antibacterial and fungicidal proteins and peptides to prevent infection, as detailed in the platelet secretome topic below.

The degranulation of activated platelets stimulates the recruitment of neutrophils and monocytes, the latter of which can differentiate into wound healing M2-macrophages at the lesion site. Macrophages are thought to be crucial for coordinating later events in response to injury; however, the importance of neutrophils and macrophages in wound repair is still debated [48,56]. As a “natural factory” of growth factors (GFs), platelets orchestrate the release and production of specific GFs, cytokines, and/or chemokines in a spatial–temporal manner that initiates the second stage of wound repair—with new tissue formation—characterized by cellular proliferation and the migration of different cell types. Among the stored GFs, essential for wound repair, is the platelet-derived growth factor (PDGF) with the -AB, -BB, and -C isoforms predominating in platelets, the transforming growth factor beta-1 and -2 (TGF-β1 and 2), bone morphogenetic proteins (BMPs), vascular endothelial growth factor (VEGF, essentially VEGF-A), basic fibroblast growth factor (bFGF, also known as FGF-2), hepatocyte growth factor (HGF), epidermal growth factor (EGF), and insulin-like growth factor-1 (IGF-1) [32,57]. These proteins are secreted by exocytosis through the fusion of granules with the plasma membrane, allowing their extracellular deposition [58].

At later stages of wound healing, fibroblasts/myofibroblasts (with actions of TGF-β and CXCL12/SDF-1) become the dominant regulatory cells. Platelet-derived factors continue to play an important role throughout this entire wound repair process, although they are most pronounced in the initiation of this complex wound healing cascade [49,59]. In the process of angiogenesis, platelets have an essential role in stimulating endothelial cells in the site vicinity in order to multiply and construct new capillaries. Some growth factors are spotlighted by a spatial–temporal gradient and synergic actions, for example, VEGF, EGF, thrombospondin-1 (TSP-1), FGF-2, IGF, and PDGF-BB while secreted from platelets; these growth factors are induced in other cells by the release of platelets, generating the autocrine and paracrine stimulation loops. This signaling stimulates the formation of new blood vessels and their maturation in the stroma underlying the wound site. In contrast to these mitogenic growth factors, TGF-β is a negative regulator of wound re-epithelialization. However, TGF-β is involved in the cellular epithelial-to-mesenchymal transition, a process thought to play a role in wound healing but also contribute to scarification [60,61]. During this stage, all processes activated after injury wind down and cease. Most of the excess endothelial cells, macrophages, and myofibroblasts undergo apoptosis (programmed cell death) or exit the wound site, leaving a mass that contains few cells and consists mostly of collagen and other extracellular matrix proteins (an unorganized network of extracellular matrix material), which ultimately forms scar tissue, also known as fibrotic tissue [49,58].

The fibrotic response detailed here not only hampers regeneration but can also be a severe medical problem unto itself. Several diseases respond to injury by creating scar tissue, which permanently and progressively may harm the functioning of many organs (such as the liver and heart). Indeed, the strategy to follow is to skew-wound repair towards regenerative capacity rather than scar formation. However, how to take that capacity and do that in humans by intervening in what is a scarring process and make it a true regenerative or restorative process? What happens in the body is that the injury is first approached by a massive number of cells (platelets) coming from around the body and going to the site of injury where they start creating scar/wound healing and fixing the damage by replacing it with scar tissue. The idea would be that perhaps we can figure out what kind of cells could give rise to native tissue. We will call them regenerative cells. Can we figure out what they are? Can we cause them to go to the site of injury? Could we inject them into someone and, thereby, start repairing tissues? This is called cell-based medicine or regenerative medicine, and its practice allows for the setting up of different routes that coordinate the events that lead to scar formation by driving down a new road through new mechanisms that might make it possible to treat a myriad of issues. For example, our immune system needs to be re-engineered by learning from the same kind of science/technology/innovation [59]. Indeed, researchers already follow these strategies, which is why regenerative medicine is considered a multi-disciplinary field that is increasingly technologically rich and knowledge-based. Regenerative medicine follows strategies that include the administration of small molecules, manipulation of the mechanical environment (for example, negative-pressure wound therapy to increase healing) or electrical environment, the use of gene-therapy approaches, cell-based strategies (including the administration of appropriate stem cells), and biomimetic scaffolds—a promising scenario for the application of biotech-educated platelets.

## 3. Creating Value: The Spark of Innovation—Reshaping the Status Quo and “Looking Beyond”

The new iconic status of platelets means that this status is often exploited as an innovation point in the intention of tissue regeneration or new product developments. Consequently, the technology or innovation behind prototypes becomes an endorsement when designing the product, mainly with therapeutic claims. For non-traditional players, such as biotech startups, new innovations reshape the status quo of already established processes such as the production of platelets in bioreactors through the technology proposed by Platelet Biogenesis [62]. The founders created an allogeneic cell therapy platform based on the production of cell therapeutics in the form of platelet-like cells (PLCs™) derived from human induced pluripotent stem cells by engineering a reliable source of platelets. In the meantime, other innovative startups, such as Mill Creek Life Science in the US and Platome Biotechnology in Iceland, have developed a human platelet lysate to accelerate regenerative medicine therapies focusing on platelet biotechnology. Other companies in Europe, such as Plasfer, have explored the power of platelets to deliver nucleic acids in the treatment of a diverse array of cancerous and noncancerous diseases. While regenerative medicine is expected to ultimately improve patient safety, in South America, PlateInnove Biotechnology is invested in the development of platelet solutions, for example, a human platelet lysate gel that contains extracellular matrix (ECM) precursors, offering a fully human option from a biotech-educated platelet platform. Platelets orchestrate complex biological responses throughout human life, providing their platelet factors in a synchronized and effective solution. The transposition of all this multifactorial and multifunctional complexity into a platform was only possible by applying the concept of biotech-educated platelets to alter the status quo in cell biology, focus on tissue repair.

## 4. Platelet Secretome (Degranulation), a “Jack-In-The-Box”—the Pivotal Process on Wound Healing-Platelet Proteome, Peptidome and Sheddome

The agonist-dependent secretion of platelet granules is a pivotal event in platelet physiology. Platelet degranulation is responsible for an effective platelet response according to microenvironment cues, including those in wound healing and tissue regeneration events. Platelet activation has long been known to be accompanied by secretion from at least six types of compartments: (1) dense granules, the primary source of small molecules [16]; (2) alpha-granules, the major protein storage organelle [63]; (3) lysosomes, the site of acid hydrolase storage [64]; (4) dense tubular systems (endoplasmic reticulum) that store and secrete protein disulfide isomerase [65]; (5) Golgi apparatus (canalicular system), storage sites of Golgi glycosyltransferases and sugar nucleotides, proteins, and small molecules that are secreted upon platelet activation [66]; (6) a new class of secretory granules, T-granules, that was recently described as containing toll-like receptor 9 [67]. Despite the presence of so many compartments, the Golgi apparatus, a dense tubular system, and T granules are not considered as the conventional compartments of platelet protein or molecule secretion.

An important point that we must highlight here is the origin of the platelets’ sub-compartment content. Platelet proteins are either inherited from their progenitor cells, the megakaryocytes [66]—absorbed from neighboring fluids, and especially plasma (platelets have been described by researchers as “hijackers/sponges” [10])—or synthesized de novo via RNA translation [68].

Several studies have been conducted highlighting the contents of these sub-compartments to try to elucidate the constitutive and inducible platelet content with OMICS tools being widely employed. It is evident that the proteomic profile of platelets differs considerably from nucleated eukaryotic cells and cultured cell lines, owing to their ontogeny as anucleate fragments of megakaryocytes and their versatile role and plasticity in the face of diverse agonists and several (patho)-physiological processes. Platelet proteomics generated some impressive numbers, with more than 5000 proteins currently identified, and more being discovered regularly. Of these, 300–350 proteins have been proven to be secreted by thrombin-activated platelets [14,16,68,69,70]. A timed cargo release is physiologically essential to the thrombus formation or its related actions (e.g., wound healing and angiogenesis) [71,72]. Changes in protein content are also common due to the shedding of proteins from the platelet surface (sheddome) and post-translational protein modifications upon the perturbation of the resting state dependent on agonist activation. Furthermore, substantial differences may be observed when comparing platelets derived from different donors, with various genomic backgrounds, or comparing healthy versus disease states. Given the variety/quantity and complexity of the data obtained by OMICS tools, it is clear, though under-evaluated in terms of physiological value, that platelets make a distinction among the “dangers” they face. They secrete discrete and optimized assortments of active biomolecules, representing the profiles of biological response modifiers.

It is now thought that the shedding of surface proteins (from activated platelets) potentially modifies platelet function and may provide a source of bioactive fragments. In theory, the proteolytic shedding of platelet membrane proteins can serve several roles, including the modulation of adhesive and cohesive interactions, limiting responses to agonists, and enabling cryptic functions of cleaved proteins [73]. These bioactive fragments can bind to receptors on other cells modifying their behavior and contributing to processes as diverse as inflammation and wound healing/tissue regeneration.

The proteolytic processing of platelet surface membrane proteins is also referred to as ectodomain shedding, and the A disintegrin and metalloproteases (ADAM) (zinc-dependent family) proteases represent one of the most diversely expressed and (patho)-physiologically relevant classes of ectodomain-shedding. ADAM-dependent ectodomain-shedding was shown to be essential for embryonic cell-differentiation and cell-fate decisions among various tissues, and homeostasis and regenerative processes in adult cells included in platelets. The proteolytic cleavage of the platelet GPIb–IX–V complex and GPVI ectodomain by metalloproteinases, such as ADAM17 and ADAM10, respectively, can negatively regulate platelet adhesion and activation. The precise molecular function of ADAM metalloproteases for the development and homeostasis of platelets is currently under exploration [17,74].

Based on spectral counts, the protein representation in the platelet sheddome varies considerably in different studies. Some studies detected semaphorin 7A, a shedded product that has been identified recently in platelets [17,74]. Semaphorin 7A, which is associated with alpha-granules, is also expressed on the platelet surface from which it may be cleaved and shed, playing critical roles in the regulation of the immune response. ADAM10 inhibitors are potential candidates to control their exacerbated activities. Several studies have now defined the subsets of membrane proteins as sheddome candidates, forming the basis for further studies examining the impact of ectodomain shedding on platelet function [74,75]. While it is clear that platelets can shed surface proteins upon ligand binding or platelet activation, for the most part, their role is not entirely understood; OMICS tools combining sheddomics and proteomics may allow for the exploration of this frontier. Plasma and platelet peptidomics have generated mass and sequence databases from which potentially interesting candidates for biotechnological exploitation for regenerative medicine may be gleaned [75,76]. In particular, the following have been identified: peptides from fibrinogen A (RGD peptides), fibrinogen B, vitronectin, desmocollin, thymosin B4 (TB4), ubiquitin, kininogen (light molecular weight chain), osteopontin, TSP-1, platelet basic protein, pigment endothelium-derived factor, angiogenin I, collagen XVIII, transforming growth factor-binding protein, IGF-binding protein, platelet factor 4 (PF4), enzymes and enzyme inhibitors (lysozyme, carboxypeptidase N, proteases and proteases inhibitors, logically coagulation factors, plasmin, and their respective inhibitors), and as mentioned before, cytokines and growth factors in high concentrations [76]. Antimicrobial peptide sequences are present in PF4, CCL-5/regulated upon activation of normal T cell expressed and secreted (RANTES), basic platelet protein, and fibrinopeptides A and B released during clotting. Another peptide found in the platelet content that has multiple functions is TB4, the main G-actin binding acetylated peptide, which acts on wound healing, stimulates angiogenesis, and suppresses inflammation. Its intracellular concentration in platelets is above 300 µM, while its concentration in serum and plasma corresponds to about 1% of the total amount of TB4 present in whole blood [75,76]. Thus, this molecule represents a biotechnological target with an excellent market potential for the treatment of chronic wounds, for example.

Indeed, platelets and their protein secretory processes, as well as peptide generation, continue to present surprises, like a “jack-in-the-box”. It is remarkable that so much of the platelet function is still poorly understood, even after 65 years of research into these cell fragments. Overall, applying our knowledge on the concept of “biotech-educated platelets” to the field of platelet research will not only result in an increased understanding of the role of platelet secretion in wound healing and tissue regeneration, but will lead to the development of usable products with a wide range of applications because of the concept’s high potential (Figure 1). There are many important and complex open questions about the extracellular regulation of tissue repair that impact the clinical practice (Table 1). Answering these questions requires the use of multiple tools, including our non-traditional biotech-educated platelets platform.

## 5. Concluding Remarks

Over the past few years, scientists have discovered many characteristics and functions of platelets, in addition to hemostasis. They are involved in: inflammation, the hematogenous dissemination of tumor cells, drug delivery systems, and tissue regeneration. The molecular exploration of platelets has been opening different avenues in terms of the application of these cells. Not surprisingly, some recent startups have been developing products derived from platelets to solve problems by using these cells’ variety of functions (Figure 2). The rational stimulation of platelets is one of the strategies, therefore, used to generate a multitude of bioactive molecules that can be used in regenerative medicine.

## Figures and Tables

**Figure 1 ijms-21-06061-f001:**
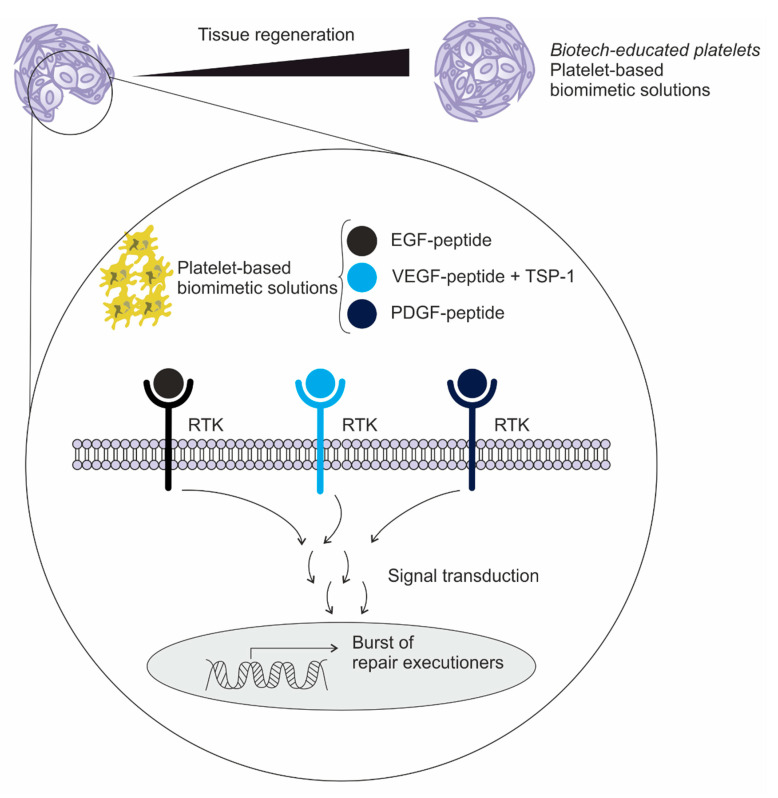
Schematic drawing of biotech-educated platelets’ growth factors or platelet-based biomimetic peptides interacting with receptor-type protein tyrosine kinase (RTK) on the cell surface of skin keratinocytes, for example, leading to coordinated efforts of several cell types including keratinocytes, fibroblasts, endothelial cells, macrophages, and more platelets. The migration, infiltration, proliferation, and differentiation of these cells will culminate in a good inflammatory response, the formation of new tissue, and ultimately tissue regeneration. This complex process is executed and regulated by an equally complex signaling network involving numerous growth factors, cytokines and chemokines. Of particular importance are: the endothelial growth factor (EGF), vascular endothelial growth factor (VEGF), thrombospondin-1 (TSP-1) and platelet-derived growth factor (PDGF) subtypes [32].

**Figure 2 ijms-21-06061-f002:**
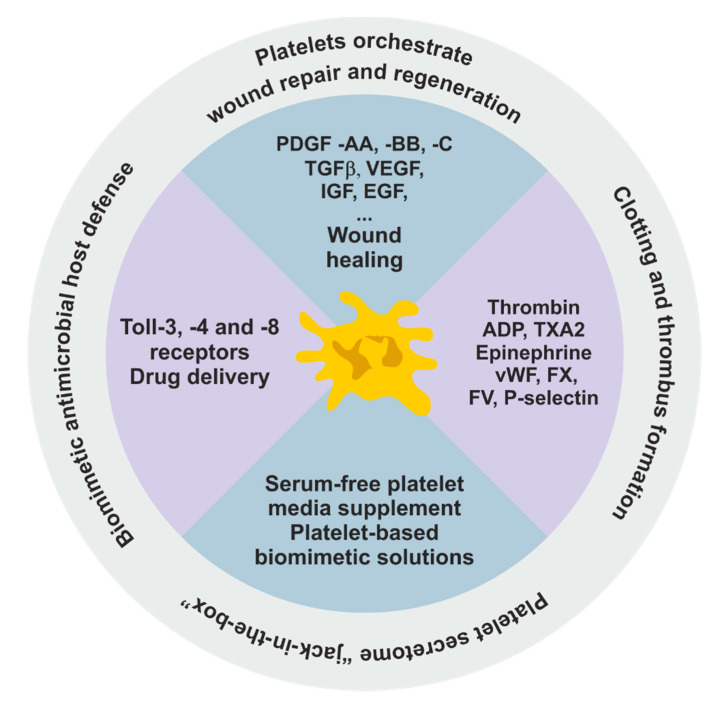
Hallmarks of platelets: Platelets, as multipurpose cells (lacking nuclei and genomic DNA), are now considered versatile cells that continue to regulate protein generation after megakaryocyte fragmentation. New discoveries have revealed that platelets are also actively involved in many physiological events beyond hemostasis and thrombosis, including angiogenesis, tissue regeneration, infection, and immunity. Some of the discoveries about platelets have emerged in the current scenario of biotechnological innovations, such as serum-free platelet media supplements, a human platelet lysate for mesenchymal stem cell expansion and platelet-based biomimetic solutions for future applications in regenerative medicine.

**Table 1 ijms-21-06061-t001:** The potential use of biotech-educated platelets in the clinical practice.

OPEN QUESTIONS	FUTURE PERSPECTIVES
What can new omics, like sheddomics, tell us about platelets’ solutions in the context of tissue repair and regeneration?	Improving the design of platelet-synthetic biosensitive materials to induce cell migration and differentiation.
How does spatiotemporal control and source of growth factors result in effective sprouting of new blood vessels followed by maturation?	Implementing sequential delivery of platelet-based solutions to match the normal signaling in diseased tissue.
How to mimic the complexity and dynamic process of wound repair and regeneration? How signals from multiple growth factor receptors are integrated for the development of a functional vascular network?	Using angiogenesis 3D models with screen combinations of cues. Determining an effective induction of therapeutic angiogenesis while minimizing side effects to promote tissue repair.

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
