# Peer review of "Biotech-Educated Platelets: Beyond Tissue Regeneration 2.0"

_ijms, 2020, doi:10.3390/ijms21176061_

Round 1
Reviewer 1 Report
Accepted for publication as the authors address all my previous comments
Author Response
I have the pleasure to resubmit the review “Biotech-educated Platelets: beyond tissue regeneration 2.0” for consideration to be published in the International Journal of Molecular Sciences. We revise the English language as suggested by the reviewers, which are addressed in the text that follows.

Reviewer 2 Report
As in my previous opinion, I recommend rejection of the manuscript.
Author Response
To Reviewer 2
International Journal of Molecular Sciences (IJMS)
Special Issue "Blood-Derived Products for Tissue Repair/Regeneration 2.0"
Manuscript ID: IJMS-851461
"As in my previous opinion, I recommend rejection of the manuscript."
A: We appreciate the reviewer for analysis of the text.
Sincerely,
Sheila Siqueira Andrade
Reviewer 3 Report
The review is well designed and written.
The paper is suitable for publication.
The topic is very interesting for the general scientists
The quality of presentation is good, for the main text and figures reported. Paper higlight an important question on wound healing.
Authors should pay attention in the textual revision.
Author Response
I have the pleasure to resubmit the review “Biotech-educated Platelets: beyond tissue regeneration 2.0” for consideration to be published in the International Journal of Molecular Sciences. We appreciate the reviewer's comments, and revise the English language as suggested, which are addressed in the text that follows.

Round 2
Reviewer 2 Report
The Authors might add a Table with detailed and specific benefits of their concept in terms of the potential use of "Biotech-educated platelets" in the clinical practice. For example, they could divide the Table into separate columns: "what is already known" and ""Future perspectives".
Author Response
To Reviewer 2
International Journal of Molecular Sciences (IJMS)
Special Issue "Blood-Derived Products for Tissue Repair/Regeneration 2.0"
Manuscript ID: IJMS-851461
I have the pleasure to resubmit the review “Biotech-educated Platelets: beyond tissue regeneration 2.0” for consideration to be published in the International Journal of Molecular Sciences. We include the Table as the reviewer's suggestion, which are addressed in the text that follows.
Looking forward to hearing from you,
Sincerely,
Sheila Siqueira Andrade
Comments Reviewer # 2
“The Authors might add a Table with detailed and specific benefits of their concept in terms of the potential use of "Biotech-educated platelets" in the clinical practice. For example, they could divide the Table into separate columns: "what is already known" and ""Future perspectives"”.
A: We appreciate the reviewer suggestion, and we included the Table - The potential use of “Biotech-educated platelets” in the clinical practice, with “Open Questions” (related about "what is already known") what refers to bottlenecks of tissue regeneration solutions, mainly related about micro-vascularization. In the second column we included "Future Perspectives", like suggested that are addressed in the text that follows.

Round 3
Reviewer 2 Report
The Authors added a Table in accordance with my last comments, which has improved the clarity of the manuscript. Also, considering both this modification and the previous corrections, the paper can be accepted in the present form.
This manuscript is a resubmission of an earlier submission. The following is a list of the peer review reports and author responses from that submission.
Round 1
Reviewer 1 Report
Review entitled on the “Biotech-educated Platelets: beyond tissue regeneration 2.0” by Andrade et al., has been reviewed. The work reviewed the new advances in platelet biology, functions, and new concepts of biotech educated platelets that connects advanced biomimetic science to platelet-based additive manufacturing for tissue regeneration. Review has completely covered the biotechnological advances and current developments in biology and functions of the stem cells. Although the work merited for the publication. It need the minor revisions in particularly gummer and spell checks. In addition, the article specifically addressed the following questions before it considered for the publication.
In the line 30-33, Why the authors mentioned the coronavirus pandemic? Since the focus of the review is biology and function of the tissue regeneration? It is mislead the concept and objective of the review.
Check line no 68….. nanorobots?
There are already incomprehensible parts in the summary which is further emphasized in other parts of the paper.
It is not possible to understand individual sentences, much less the meaning of individual parts. In addition, there are many typographical errors in the paper. Even in citing the literature, there are errors.
Moreover, the although the authors provided some pictures, it is not sufficient to highlight the current status of the biology and function of the platelets. Therefore, it the author provided the additional table or figures with fundamental development on platelet biology is good for readers.
In line 483. Development stages need to be addressed why it is called as tissue regeneration 2.0.? then what is the tissue regeneration 1.0? explain development cycle from 0-2.0?
The authors not provided the conclusion portion in the article with take home message.
Overall the article need the minor revision, particularly addressing the fundamental of tissue regeneration and platelets biology and function in connection with current development. It give the idea to new researchers.
Author Response
To Editors of International Journal of Molecular Sciences (IJMS)
Special Issue "Blood-Derived Products for Tissue Repair/Regeneration 2.0"
Manuscript ID: IJMS-851461
I have the pleasure to resubmit the review “Biotech-educated Platelets: beyond tissue regeneration 2.0” for consideration to be published in the International Journal of Molecular Sciences. We considered all comments and modifications suggested by the reviewers, which are addressed in the text that follows.
Looking forward to hearing from you,
Sincerely,
Sheila Siqueira Andrade
#Reviewer 1#
Review entitled on the “Biotech-educated Platelets: beyond tissue regeneration 2.0” by Andrade et al., has been reviewed. The work reviewed the new advances in platelet biology, functions, and new concepts of biotech-educated platelets that connects advanced biomimetic science to platelet-based additive manufacturing for tissue regeneration. Review has completely covered the biotechnological advances and current developments in biology and functions of the stem cells. Although the work merited for the publication. It need the minor revisions in particularly gummer and spell checks. In addition, the article specifically addressed the following questions before it considered for the publication.
In the line 30-33, Why the authors mentioned the coronavirus pandemic? Since the focus of the review is biology and function of the tissue regeneration? It is mislead the concept and objective of the review.
A: We removed this information (about coronavirus on context of Biotechnology) of the review, as reviewer suggestion.
We cannot fail to mention our current and most dramatic problem with the coronavirus pandemic, a scenario that is already demanding the most from the ongoing and future biotechnologies available, with the development of new drugs and vaccines, in addition to quickly and efficiently diagnostics’ approaches.
Check line no 68….. nanorobots?
A: Dear reviewer, we added more information about the nanorobots in the text, Esteban-Fernández de Ávila et al., 2018 (Science Robotics) report the integration of diverse biological functions from the plasma membranes of two cell types, red blood cells and platelets, into a single nanorobot surface to create a robust biomimetic nanorobot for multipurpose biodetoxification and concurrent removal of pathogenic bacteria and toxins in particular.
Pag 3. …(3) production of biomimetic nanorobots consisting of gold nanowires cloaked with a hybrid of red blood cell membranes and platelet membranes for removal of pathogenic bacteria and toxins [22];…
There are already incomprehensible parts in the summary which is further emphasized in other parts of the paper.
A: Dear reviewer, we analyzed the summary/abstract, and included the reference Sipe et al., 2004 (“Localization of bone morphogenetic proteins (BMPs)-2, -4, and -6 within megakaryocytes and platelets”, Bone Journal) listing another factor (BMP) to the process of morphogenesis.
Page 7…Among the stored GFs essential for wound repair is the platelet-derived growth factor (PDGF) with the -AB, -BB, and -C isoforms predominating in platelets, transforming growth factor beta-1 and -2 (TGF-β1 and 2), bone morphogenetic proteins (BMPs), vascular endothelial growth factor (VEGF, essentially VEGF-A), basic fibroblast growth factor (bFGF, also known as FGF-2), hepatocyte growth factor (HGF), epidermal growth factor (EGF), and insulin-like growth factor-1 (IGF-1) [33,64].
In addition to BMP, in the text we mention several known events related to morphogenesis with the healing and tissue repair process (topic 2. Unexpected Platelets Biochemistry Orchestrating the entire process of Wound Repair and Tissue Regeneration).
It is not possible to understand individual sentences, much less the meaning of individual parts. In addition, there are many typographical errors in the paper. Even in citing the literature, there are errors.
A: Dear reviewer, we check point by point/line by line in the review and typographical and individual sentences were revised in the text.
Moreover, the although the authors provided some pictures, it is not sufficient to highlight the current status of the biology and function of the platelets. Therefore, it the author provided the additional table or figures with fundamental development on platelet biology is good for readers.
A: We included the figure as suggested.
Page 17
Figure 4. Schematic drawing of Biotech-educated platelets, growth factors or platelet-based biomimetic peptides interact with receptor-type protein tyrosine kinase (RTK) in cell surface of skin keratinocytes for example, leading to a coordinated efforts of several cell types including keratinocytes, fibroblasts, endothelial cells, macrophages, and more platelets. The migration, infiltration, proliferation, and differentiation of these cells will culminate in an (good) inflammatory response, the formation of new tissue and ultimately tissue regeneration. This complex process is executed and regulated by an equally complex signaling network involving numerous growth factors, cytokines and chemokines. Of particular importance are: EGF, VEGF, TSP-1 and PDGF subtypes [33].
In line 483. Development stages need to be addressed why it is called as tissue regeneration 2.0.? then what is the tissue regeneration 1.0? explain development cycle from 0-2.0?
A: Dear reviewer, we follow the context of Special Issue of this journal "Blood-Derived Products for Tissue Repair/Regeneration 2.0", and in the last section “5.Translational Approaches for Biotech-educated platelets” we discussion the evolution of tissue regeneration 1.0 to 2.0. We included a brief mention of the repair 1.0 and tissue regeneration 2.0. In addition, it’s important to stress, the review focused on discussing platelets in the context of tissue regeneration, and the convergence between synthetic biomaterials (repair 1.0) and cell-based medicine in tissue regeneration (tissue regeneration 2.0).
Page 18… An evolution and natural convergence between tissue repair 1.0 and tissue regeneration 2.0. This platelet platform allows taking advantage of the plasticity of these cells, delivering valuable benefits to human health and the life science market through a new vision of tissue regeneration 2.0.
The authors not provided the conclusion portion in the article with take home message.
A: We appreciate the suggestion and believe that the inclusion of the above excerpt; the conclusion of the text is satisfactory.
Overall the article need the minor revision, particularly addressing the fundamental of tissue regeneration and platelets biology and function in connection with current development. It give the idea to new researchers.
A: Dear reviewer, we try to do that.

Reviewer 2 Report
The manuscript is not focused on any specific issue related to platelets. The text is full of very general and unproven considerations and - in my opinion - would have a limited value for the readers of the Journal.
In partcular, section No. 5. entitled "Translational Approaches for Biotech-educated platelets" does not contain virtually anything relevant despite the fact that the manuscript title is: "Biotech-educated Platelets: beyond tissue 2 regeneration 2.0". Actually, I was unable to detect any evidence of even 1.0 version in the text.
Author Response
To Editors of International Journal of Molecular Sciences (IJMS)
Special Issue "Blood-Derived Products for Tissue Repair/Regeneration 2.0"
Manuscript ID: IJMS-851461
I have the pleasure to resubmit the review “Biotech-educated Platelets: beyond tissue regeneration 2.0” for consideration to be published in the International Journal of Molecular Sciences. We considered all comments and modifications suggested by the reviewers, which are addressed in the text that follows.
Looking forward to hearing from you,
Sincerely,
Sheila Siqueira Andrade
#Reviewer 2#
The manuscript is not focused on any specific issue related to platelets. The text is full of very general and unproven considerations and - in my opinion - would have a limited value for the readers of the Journal.
In partcular, section No. 5. entitled "Translational Approaches for Biotech-educated platelets" does not contain virtually anything relevant despite the fact that the manuscript title is: "Biotech-educated Platelets: beyond tissue 2 regeneration 2.0". Actually, I was unable to detect any evidence of even 1.0 version in the text.
A: We appreciate the reviewer for analysis of the text. We try to transmit the high innovations in area of platelet function in the context of tissue regeneration, as well as reviewing the convergence between a biomaterials area (repair 1.0) and a cell-based medicine (tissue regeneration 2.0) in the context of special tissue Special Issue "Blood-Derived Products for Tissue Repair/Regeneration 2.0" /IJMS. In addition, the section 5 "Translational Approaches for Biotech-educated platelets" discusses precisely the cycle and integration from tissue regeneration 1.0 to tissue regeneration 2.0.
Round 2
Reviewer 2 Report
The manuscript still has considerable limitations which I mentioned in the first review round. Importantly, a large amount of very general considerations and low interest for the Readers constitute its major drawbacks.